# Understanding Redundancy in Discrete Multi-Agent Communication

**Jonathan D. Thomas**
University of Bristol
jt17591@bristol.ac.uk

**Raúl Santos-Rodríguez**
University of Bristol
enrsr@bristol.ac.uk

**Robert Piechocki**
University of Bristol
r.j.piechocki@bristol.ac.uk

## Abstract

By providing agents with the capacity to learn sample-efficient and generalisable communications protocols, we may enable them to more effectively cooperate in real-world tasks. In this paper, we consider this in the context of discrete decentralised multi-agent reinforcement learning to provide insights into the impact of the often overlooked size of the message set. Within a referential game, we find that over-provisioning the message set size leads to improved sample efficiency, but that these policies tend to maintain a high-degree of redundancy, often utilising multiple messages to refer to each label in the dataset. We hypothesise that the additional redundancy within these converged policies may have implications for generalisation and experiment with methodologies to gradually reduce redundancy while maintaining sample efficiency. To this end, we propose a linearly-scheduled entropy regulariser which encourages an agent to initially maximise the utilisation of the available messages but, as training progresses, this incentive is removed. Through this mechanism, we achieve improved sample efficiency whilst converging to a model with significantly reduced redundancy and that generalises more effectively to previously unseen data.

## 1 Introduction

Intuitively, in most cooperative tasks agents can benefit from communicating. The ability to interact allows them to share information with their collaborators, facilitating better cooperative decisions. Through studying the emergent properties of this type of communication, we hope to equip cooperative agents with the necessary tools to quickly develop protocols which generalise to unseen states.

Within this work, we focus on the emergence of discrete communication protocols in Multi-agent Reinforcement Learning (MARL) with a particular interest in Independent Learners (IL) [1]. This challenging domain has previously been considered within a number of previous works [2, 3, 4]. However, little attention has been paid to the impact of the dimensionality of the message set on the messaging protocol, which could lead to unexpected behaviours. As an example, we explore this within an MNIST-based referential game (shown in Figure 1). The game requires two agents, a speaker and a listener, to derive a protocol to facilitate communication of the speaker's observed digit to the listener via a set of discrete messages, $M$. We experiment with message set sizes of $10, 20, 30$ and $40$. The utilisation of MNIST [5] should mean that $10$ messages are sufficient, but we find that over-allocation improves sample efficiency on training data. Interestingly the derived policies tend to retain a high-degree of redundancy and typically utilise multiple messages to refer to the same digit.

We further build upon this result and investigate methods to reduce the redundancy within the protocol. By doing so, we hope to build more robust representations that better capture distinct concepts that a speaker (or agent more generally) may wish to communicate. We achieve this through the introduction of a linearly-scheduled entropy regulariser into the speaker's loss function. At the beginning of training, this term acts to maximise the utilisation of the message set which has previously been

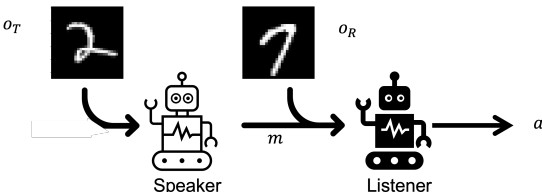

Figure 1: MNIST-based Referential Game comprised of two agents, a speaker and a listener.

demonstrated to improve performance in this setting [4]. As training progresses, we linearly reduce this incentive until it is zeroed out. Once the incentivise is removed, the speaker tends towards a policy which reduces its utilisation of the message set, thereby achieving our desired reduction in effective cardinality. We empirically validate this approach and observe an improvement in sample efficiency and generalisation.

## 2   Related Work

Enabling communication among agents has attracted significant attention within the multi-agent reinforcement learning (MARL) literature [6]. Several methods have been proposed including those that utilise graph neural networks [7, 8] and other forms of differentiable communication channels [2]. Within this work, we focus on discrete communication channels and independent learners (IL) [1] which has been previously considered by [3, 4]. The use of IL prohibits differentiable communication channels and parameter sharing among agents which makes learning more difficult but is considered to be a more biologically plausible method for learning. It also easily permits extension to situations where a communicating agent may not be differentiable. Our work differs from existing work as it explores the impact of message dimensionality within discrete communication in MARL and experiments with methodologies to reduce excessive redundancy.

## 3   Method

In Section 3.1 we define our setting. As the aim of this work is to investigate the impact of message set size in MARL, in Section 3.2, we demonstrate that over-provisioning $M$ results in improved sample-efficiency, but that the policies tend to maintain a higher than necessary level of redundancy in their messaging protocols. We then propose a methodology to address this problem in Section 3.3.

### 3.1   Setting

We apply the MARL approach defined in this paper to an $N$-player partially observable stochastic game [9]. This is defined by the tuple $G = \{\mathcal{I}, \mathcal{S}, \{A^i\}, \{M^i\}, \{O^i\}, \mathcal{P}, \Omega, \{R_i\}\}$. Where, $\mathcal{I}$ is a finite set of agents indexed by $\{i, \ldots, N\}$, $\mathcal{S}$ is the state space, $A^i$ and $M^i$ represent the finite set of actions and messages that are available to agent $i$. We refer to the joint action and message space as $\mathcal{A} = A^1 \times \ldots \times A^n$ and $\mathcal{M} = M^1 \times \ldots \times M^n$. At each time-step agent $i$, receives a partial observation $o^i \in O^i$ which is defined by the observation function $\Omega : \mathcal{S} \times \mathcal{I} \to O$. Additionally, all agents also receive all messages sent by other agents in the previous time-step which we refer to as $m^{-i} \in \mathcal{M}^{-i}$. Agents use this information to select an action $a^i \in A^i$ according to $\pi_{i,a}$ and a discrete message $m \in M_i$ according to the policy, $\pi_{i,m}$. They then experience a state transition $T : S, \mathcal{A} \to S$ and receive a reward $R^i : S, \mathcal{A} \to \mathcal{R}$. Agents are tasked with finding action policies $\pi_{i,a} : (o, m^{-i}) \to A_i$ and a message policy $\pi_{i,m} : (o, m^{-i}) \to M_i$ which maximise the cumulative discounted reward they receive.

### 3.2   Problem Demonstration

We demonstrate the problem in an MNIST-based referential game shown in Figure 1. A referential game consists of two agents a speaker, $s$, and a listener, $l$. The speaker, $s$, observes $O^s$ and the listener, $l$, observes $O^l$. The observations are sampled from the MNIST dataset. The agents undertake a cooperative task that requires the listener to sum the digits which the observations represent. If

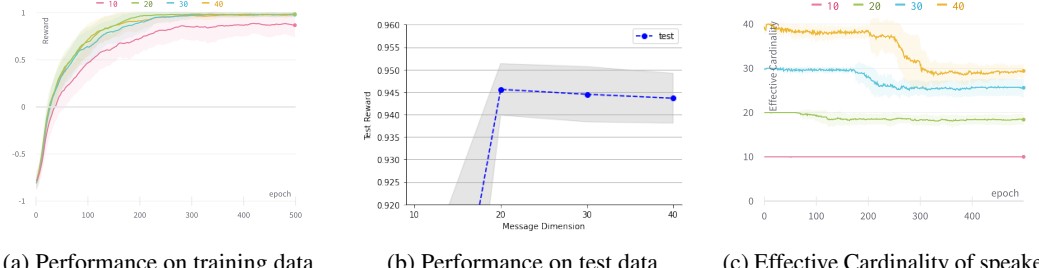

(a) Performance on training data     (b) Performance on test data     (c) Effective Cardinality of speaker

Figure 2: Figures showing the performance of message set cardinalities $|M| = [10, 20, 30, 40]$. Plotted results show mean and $95\%$ confidence interval.

the correct answer is given both agents get a reward of $1$, otherwise they get a reward of $-1$. The only mechanism to achieve this is through effective utilisation of the uni-directional communications channel which connects $s$ to $l$. The channel permits $s$ to transmit a single-message $m^s \in M^s$ to $l$. Successful completion of this task requires $s$ and $l$ to agree on a mapping from $O^s \rightarrow M$ and for the listener to be able to identify $O^l$. We instantiate the speaker, $\pi_{s,\theta}$, and listener, $\pi_{l,\phi}$ as deep neural networks. Both agents are trained independently utilising REINFORCE [10]. We additionally utilise speaker and listener biases described by [4] to promote the emergence of communication. We conduct experiments with 10, 20, 30 and 40 messages each repeated with 10 random seeds. We use the same hyperparamaters defined within [4] and implement our code in PyTorch Lightning [11][1].

The performance of different cardinalities of $M$ are shown in Figure 2. It is apparent from Figure 2a that over-provisioning of $M$ results in improved sample efficiency. $|M| = 10$ is significantly less effective and does not converge to a policy which is able to reliably communicate the speaker's observation achieving an average test reward of $0.84$. Over-allocation of $M$ may afford a degree of flexibility within learning and reduce the difficulty of the task. The extra capacity could allow the speaker to represent more distinct members of an image class as separate messages, whereas $M = 10$ requires the speaker to learn that these images represent the same concept. This is supported by Figure 2c, which shows the *effective cardinality* of the message protocol. We define *effective cardinality* in eq. (1) which tells us which messages are being used by the speaker on a large batch of validation data, $B$. It illustrates how speakers tend to converge to messaging protocols which utilise a subset of $M$ which is larger than the minimum viable size of 10. We expected that this may manifest in a reduction in test performance – this is supported by Figure 2b which shows a negative correlation between test performance and the remaining cardinality from $|M| \geq 20$.

$$c(B) = \sum_{m \in M} \mathbb{1}[iffm \in B] \tag{1}$$

### 3.3 Scheduling Entropy Regularisation

By addressing the excessive redundancy in the speaker's converged messaging protocols, we hope to improve generalisation. We consider the speaker-side bias proposed within [4] as our starting point and further build upon them. In their standard form, these act to incentivise a speaker to maximise the mutual information between messages and observations. Practically, this is implemented into the speaker's loss through Equation (2). Where $\overline{\pi^i_M}$ is the average of the message policy over a batch of experience, and, $\lambda$ and $\mathcal{H}_{target}$ are both hyperparameters. This encourages the speaker to output messages with uniform probability on average, but are not random with respect to an observation.

$$L_{ps}(\pi^i_M, o^i) = -\mathbb{E}(\lambda \mathcal{H}(\overline{\pi^i_M}) - (\mathcal{H}(m^i_t | o_t) - \mathcal{H}_{target})^2) \tag{2}$$

We speculate that maximisation of $\mathcal{H}(\overline{\pi^i_M})$ encourages the additional redundancy as it may provide a mechanism to encourage the speaker to represent more distinct members of an image class as separate messages. As training progresses and the speaker's competence begins to increase, we

---

[1]Code available at `https://github.com/Jon17591/redundancy_reg`.

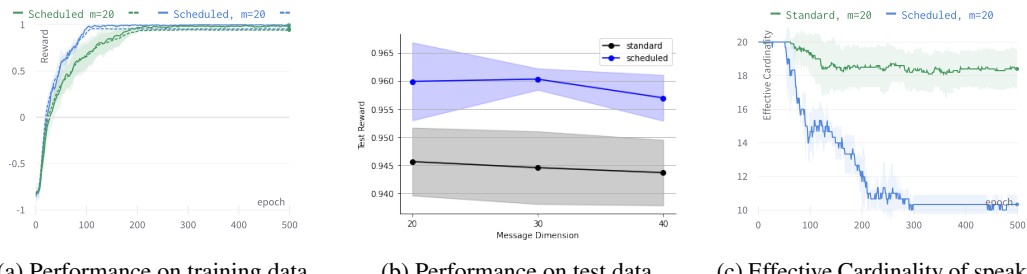

(a) Performance on training data    (b) Performance on test data    (c) Effective Cardinality of speaker

Figure 3: Figures showing the performances of the scheduling method introduced in Section 3.3. Plotted results show mean and $95\%$ confidence interval.

believe that it may be desirable to reduce the utilisation of the message set. We achieve this by modifying the behaviour of $\lambda$ and making it a function of the training epoch which we use as an estimation of the speaker's experience as shown in Equation (3). Where $t$ is specified in epochs and $c$ is a hyperparameter which controls the rate at which we reduce the entropy maximisation incentive.

$$\lambda(t) = \lambda \times max \left( \left[ 1 - \left( \frac{t}{c} \right), 0 \right] \right) \tag{3}$$

## 4    Experiments and Results

We repeat the same experimentation in Section 3 with minor modifications to our code base to facilitate the scheduling given in Equation (3). We run 3 random seeds for $500$ epochs and conduct a grid-search across $|M| = [20, 30, 40]$ and $c = [100, 200, 300]$. Our results are shown in Figure 3. We report our reward and effective cardinality during training in Figure 3a and 3c for our best-performing parameter combination which was $|M| = 20$ and $c = 300$. These plots demonstrate an improvement in sample efficiency and that $\lambda(t)$ can reduce the effective cardinality compared to the standard implementation. Figure 3b shows the best test reward we obtained for each $|M|$, where the $c$ utilised was $c = 300$ for $|M| = 20$, $c = 200$ for $|M| = 30$, $c = 200$ for $|M| = 40$. This shows that $\lambda(t)$ can improve generalisation to unseen examples. Training and effective cardinalities plots for $c = 200, M = 30$ and $c = 200, M = 40$ are provided in Appendix B.1. We additionally provide an analysis of an alternative parameterisation of the scheduling method in Appendix B.3.

## 5    Discussion and Future Work

Our results support our hypothesis that addressing excessive redundancy can improve generalisation performance as shown in Figure 3b. The scheduled entropy regularisation, $\lambda(t)$, facilitates this and can be shown to significantly reduce the excess redundancy in the protocol as $\lambda(t)$ tends to $0$ which is shown in Figure 3c. We believe its introduction encourages the speaker to learn more robust representations that better capture distinct concepts which manifests as an improvement in test performance. The improvement in sample efficiency is somewhat surprising and we hypothesise that it may suggest that maximisation of $\mathcal{H}(\overline{\pi_M^i})$ may be counter-productive as the model converges. We further investigate this in Appendix B.2 and demonstrate empirical evidence which appears to support this hypothesis.

We believe that this method represents an interesting step towards generalisation within communicating MARL. Our current investigations were restricted to a referential game due to it being relatively amenable to analysis. Most usefully, it allowed us to know the minimal number of messages required for the task and for the agents' to concentrate on communication without distraction by other environmental tasks. Our future work intends to extend our experimentation to more complex temporally-extended environments where communications protocols may be more abstract.

## Acknowledgments and Disclosure of Funding

This work is funded by the Next-Generation Converged Digital Infrastructure (NG-CDI) Project, supported by BT and Engineering and Physical Sciences Research Council (EPSRC), Grant ref. EP/R004935/1. RSR is partially funded by the UKRI Turing AI Fellowship EP/V024817/1.

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

# A Appendix A: Further Experimental Details

## A.1 Hyperparameters

Both speaker and listener policy networks ($\pi_{s,\theta}$ and $\pi_{l,\phi}$) follow the same neural network architecture defined within [4]. This consists of 2 convolutional layers of 32 and 64 channels respectively, utilising a kernel size of 5 and a stride of 1. Both of which are followed by max-pooling. This is followed by a single fully-connected layer containing 1024 neurons. All layers use ReLU activations. All hyperparameters are kept consistent with [4], where both models are optimised using Adam [12] with a learning rate of 0.0003.

# B Appendix B: Further Experimentatal Results

## B.1 Additional training reward and effective cardinality plots

Here we show the results we omitted in Figure 3. Figure 4 and 5 show the best obtained performances for $|M| = 30$ and $|M| = 40$, respectively. In addition to the improvement in generalisation which we demonstrate in Figure 3b, we note a small improvement in sample-efficiency in Figure 4a and 5a, and a reduction in effective cardinality in Figure 4b and 5b to approximately 10.

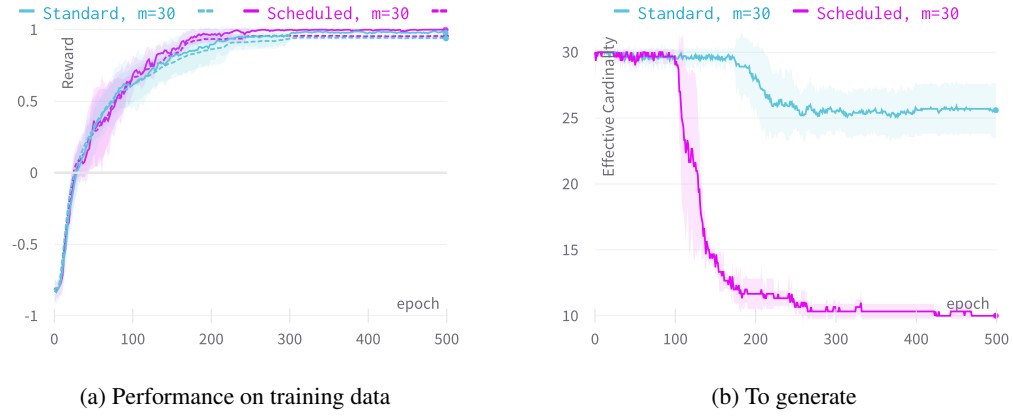

(a) Performance on training data                          (b) To generate

Figure 4: Figures showing the impact of entropy maximisation. Plotted results show mean and 95% confidence interval for message dimension of 20.

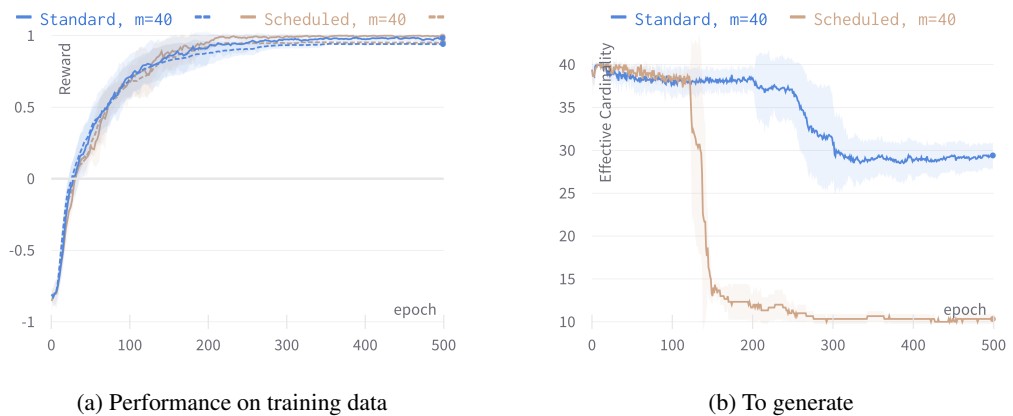

(a) Performance on training data                          (b) To generate

Figure 5: Figures showing the impact of entropy maximisation. Plotted results show mean and 95% confidence interval for message dimension of 20.

## B.2 Is this a function of the underlying algorithm?

A question that presents itself is whether the characteristics displayed within Figure 2 are just a manifestation of the speaker side biases within [4].

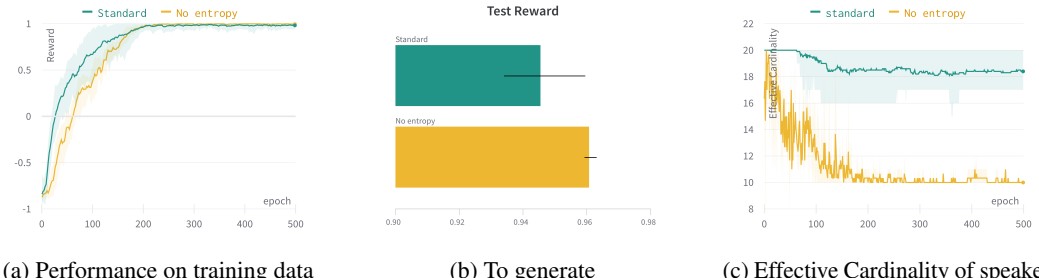

| (a) Performance on training data | (b) To generate | (c) Effective Cardinality of speaker |

Figure 6: Figures showing the impact of entropy maximisation. Plotted results show mean and 95% confidence interval for message dimension of 20.

## B.3 Explicit entropy minimisation

In our current work, we allow for entropy minimisation to happen organically. This has been shown to happen in [13]. An alternative formulation could involve explicit minimisation of entropy. Functionally, we test the implications of this through modification of eq. (3) to eq. (4). We demonstrate results which compare against Equation (3) in Figure 7.

$$\lambda(t) = \lambda \times max\left(\left[1 - \left(\frac{t}{c}\right), -1\right]\right) \tag{4}$$

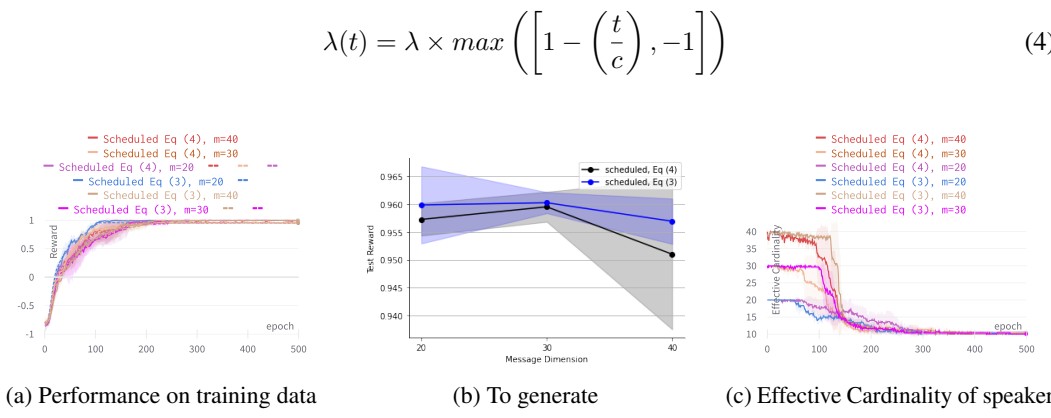

| (a) Performance on training data | (b) To generate | (c) Effective Cardinality of speaker |

Figure 7: Figures showing performance of different scheduling methods. Plotted results show mean and 95% confidence interval.

We find that this modification results in mostly comparable performance. There would appear to be a small reduction in mean test performance Figure 7b, however, it is not clear whether this is statistically significant and further analysis is required. As shown in Figure 7a, we were not able to achieve the same level of sample efficiency with the modified scheduling. Interestingly, the impact on effective cardinality is mixed which is shown in Figure 7c. For both $|M| = 30$ and $|M| = 40$ the reduction happens more quickly which is what we would have expected to happen.

