# OpenReview forum: "Understanding Redundancy in Discrete Multi-Agent Communication"
_NeurIPS.cc/2022/Workshop/LaReL — LaReL 2022_

### Official Review · Reviewer_EMMK · 2022-10-09
**Straightforward paper but results could be more compelling and additional investigations/explications could have been provided.**

**Rating:** 5
**Confidence:** 4

**Review:**

## Summary
This paper investigates the impact of the vocabulary set size for discrete communication in Multi-agent Reinforcement Learning with Independent Learners. Authors shows that a scheduled entropy regularization (first maximization, then minimization) improves test performance without impacting sample efficiency of training asymptotic performance for over-parameterized vocabularies.

## Pros
* Paper is easy to read and understand. It is clear in what it aims at investigating.
* the proposed method is simple to implement and reduces the redundancy as well as improving test performance while not hurting sample efficiency.

## Cons
* results are not very compelling:
    * l. 90-91: it is not very clear that the sample efficiency is improved as the asymptotic performance is also improved when using larger M and that 95% confidence intervals overlap.
    * l. 100-101 the test performance for M=20, 30, 40 are all within the same 95% confidence interval. Additionally, the test performance for M=10 is significantly smaller. This goes against the statement that redundancy hurts test performance.


* the methodology and experimental setup could be made clearer:
    * [4] should also be cited for the MNIST-based referential game.
    * even if this paper follows the methodology and method of [4] it should mention the used biases, the neural network architectures, etc.
* Some investigation/experiments are missing
    * The reasons for lower asymptotic training performance with M=10 are not investigated. Why is the test performance lower if there is no redundancy?
    * it is unclear from l. 88 whether the "no-scheduling" experiments uses the entropy maximization of [4]. Yet it appears to use it since in Figure 3. (c) the "scheduling" experiments experience the drop in cardinality first but still with t smaller than c. This brings the following question for the "no-scheduling" experiment where there is only entropy maximization: why is the effective cardinality smaller than the vocabulary set size and why does it drops with learning even though the asymptotic performance is already reached?
    * I feel that the experiment (entropy regularization + vocabulary size of 10) is missing. It would be interesting to see if the vocabulary size still has an effect once the regularization is used.
    * what happens if there is only entropy minimization?

## Typos/Suggestions
* Foundational papers such as *Luc Steels. A self-organizing spatial vocabulary. Artificial life 1995* should be added in the related work section.
* l.60: "demonstrating" --> demonstrate?
* l.88 "hyperparamaters" --> hyperparameters?
* l.115 concretely it is not exactly a function of the speaker's experience or performance/skill but rather of the training epoch
* l. 135 "This view may elude to parallels" to reformulate?

---

### Official Review · Reviewer_aPp5 · 2022-10-14

**Rating:** 7
**Confidence:** 4

**Review:**

Quality:
The paper thoroughly investigates the question of how excess message capacity (dimensionality of the message set) affects generalization in multi-agent communication in a referential game.

One issue with the results: does overutilization of the message capacity happen if you don't use the mutual information objective from Eccles et al? This would be good to clarify.

Clarity: The hypothesis and motivation are clearly spelled out, the method is clear, and the results crisply address the stated questions/aims. A couple small issues:
- Line 78 appears to state that the listener observes O^s, which would not make sense.
- Line 48 -> agent's -> agents
- Equation 1 is weirdly placed (put it after it is introduced).
- There appears to be a plotting error in Figure 2b. As it currently stands, it does not appear to support the claim that increased vocab size leads to worse generalization, since the line starts at the bottom of the plot indicating that generalization is very poor for a message size of < 20. What is going on with this fig?

Significance: The biggest issue with the paper is its significance. Message vocab size seems like a relatively minor hyperparameter in the communication setting. I would be more convinced of the importance, except that the effect on generalization does not appear to be significant. According to figures 2b and 3b, the decrease in test performance with a larger message dimension is very minor (from just above 0.945 test reward to just below). However, this is tempered by the fact that the proposed objective does lead to higher performance in Figure 3b.

Originality: I agree with the paper's claim that little attention has been previously been paid to vocab size in emergent communication.

---

### Decision · Program_Chairs · 2022-10-21

Accept